# Inference of Breed Structure in Farm Animals: Empirical Comparison between SNP and Microsatellite Performance

**DOI:** 10.3390/genes11010057

**Published:** 2020-01-04

**Authors:** Abbas Laoun, Sahraoui Harkat, Mohamed Lafri, Semir Bechir Suheil Gaouar, Ibrahim Belabdi, Elena Ciani, Maarten De Groot, Véronique Blanquet, Gregoire Leroy, Xavier Rognon, Anne Da Silva

**Affiliations:** 1Université de Djelfa, Djelfa 17000, Algeria; laounabbes@gmail.com; 2Laboratory of Exploration and Valorization of Steppic Ecosystems (EVES), University of Djelfa, BP 3117, Djelfa 17000, Algeria; 3Science Veterinary Institute, University of Blida, BP 270, Blida 09000, Algeria; sahraoui_vet@yahoo.fr (S.H.); medlaffri@yahoo.fr (M.L.); ibrahimveto2@gmail.com (I.B.); 4Laboratory of Biotechnology related to Animal Reproduction (LBRA), University of Blida, BP 270, Blida 09000, Algeria; 5Laboratory of Physiopathologie et biochimie de la Nutrition (PpBioNut) Department of Biology, Abou Bekr Belkaid Tlemcen University, Tlemcen 13000, Algeria; suheilgaouar@gmail.com; 6Department of Biosciences, Biotechnologies and Biopharmaceutics, University of Bari, 74100 Bari, Italy; elenaciani@biologia.uniba.it; 7Dr. Van Haeringen Laboratorium BV, Agro Business Park 100, 6708PW Wageningen, The Netherlands; mgr@molg3n.com; 8Univ. Limoges, INRAE, EA7500, USC1061 GAMAA, F-87000 Limoges, France; veronique.blanquet@unilim.fr; 9Université Paris-Saclay, INRA, AgroParisTech, GABI 78350 Jouy-en-Josas, France; gregoire.leroy@agroparistech.fr (G.L.); xavier.rognon@agroparistech.fr (X.R.)

**Keywords:** livestock diversity, cross-breeding, simple sequence repeat, short tandem repeat, single nucleotide polymorphism

## Abstract

Knowledge of population structure is essential to improve the management and conservation of farm animal genetic resources. Microsatellites, which have long been popular for this type of analysis, are more and more neglected in favor of whole-genome single nucleotide polymorphism (SNP) chips that are now available for the main farmed animal species. In this study, we compared genetic patterns derived from microsatellites to that inferred by SNPs, considering three pairs of datasets of sheep and cattle. Population genetic differentiation analyses (Fixation index, F_ST_), as well as STRUCTURE analyses showed a very strong consistency between the two types of markers. Microsatellites gave pictures that were largely concordant with SNPs, although less accurate. The best concordance was found in the most complex dataset, which included 17 French sheep breeds (with a Pearson correlation coefficient of 0.95 considering the 136 values of pairwise F_ST_, obtained with both types of markers). The use of microsatellites reduces the cost and the related analyses do not require specific computer equipment (i.e., information technology (IT) infrastructure able to provide adequate computing and storage capacity). Therefore, this tool may still be a very appropriate solution to evaluate, in a first stage, the general state of livestock at national scales. At a time when local breeds are disappearing at an alarming rate, it is urgent to improve our knowledge of them, in particular by promoting tools accessible to the greatest number.

## 1. Introduction

According to the “Second Report on the State of the World’s Animal Genetic Resources for Food and Agriculture” [1], indiscriminate cross-breeding (i.e., breeding not carried out within the framework of selection plans) and the increased use of imported exotic breeds have been identified as the main causes leading to the genetic erosion of livestock. These practices directly threaten the genetic integrity of local breeds. Developing countries, under increasing economic pressure, are the most exposed to this phenomenon. Moreover, in these countries, locally adapted breeds are still mainly reared under traditional practices and generally there is less formal organization of the livestock sector. Thus, they are made more vulnerable. In addition, the ease of herds’ mobility, which became widespread in the age of motorization, has strengthened gene flows between populations. As emphasized by Sponenberg et al. [2], gene flows, until recently naturally limited by geographical, historical and cultural conditions, are intensifying, which endangers the genetic integrity of “traditional” breeds while “standardized” and “industrial” breeds are artificially maintained in a state of protective isolation.

In such a context, knowledge of population structure appears to be essential for managing and conserving Farm Animal Genetic Resources (FAnGR). This is particularly important in developing countries, where local breeds adapted to environment, disease and social uses, represent an invaluable reservoir of genetic diversity that is threatened by agricultural modernization.

In the last few decades, microsatellites (also called short tandem repeats, (STRs) or simple sequences repeats (SSRs)) have been the marker of choice in animal population genetics. In recent years, the use of single nucleotide polymorphisms (SNPs) has become increasingly important in this area. Nowadays, advances in high-throughput sequencing have identified thousands of SNPs, leading to the development of whole-genome SNP chips for the major farm animal species [3]. Illumina Inc. provides public and commercial SNP chips for cattle (BovineSNP50v2) [4], sheep (OvineSNP50) [5], goat (GoatSNP50) [6], chicken (chip with 57,636 SNPs) [7] and pig (PorcineSNP60) [8]. Faced with the power of chips, microsatellites are increasingly left aside by research focused on genome diversity. At this time of technological progress, the major concern in the Farm Animal Genetic Resources (FAnGR) field seems to be the management of the “transition” from microsatellite data to SNP data [3]. However, the use of array technology implies a higher cost as well as the availability of IT infrastructure able to provide adequate computing and storage capacity, and indeed the staff to maintain and operate these platforms. All these conditions can be difficult for developing countries to meet.

A significant amount of work has been dedicated to the comparison between microsatellites and SNPs considering animal and plant models. If we limit the overview to vertebrates alone, we can distinguish different types of research. Some studies have compared the effectiveness of the two markers: (i) for the purpose of performing kinship analyses [9,10,11,12,13], (ii) to infer diversity parameters e.g., [14,15], and (iii) to analyze genetic relationships between populations [16,17,18,19,20]. This last group is of particular interest for this study as it aims to enrich knowledge about the precise issue of the inference of population genetic structure. The studies [16,17,18,19,20] carried out comparative analyses using a few hundred SNPs, mainly due to the limited availability of SNPs markers for the models considered at the time of the investigation. The study by Gärke et al. [18] is an exception as it considers 29 microsatellites and 2931 SNPs in chicken, with the aim of determining the number of SNPs required to achieve the same differentiation power as for a given standard set of microsatellites.

In this study, we consider the case of farm animals for which the amount of SNP available is now very high. The question asked is whether a standard set of microsatellites can provide a picture of the population genetic structure that is comparable to the pattern deduced from a set of several tens of thousands of SNPs. From this perspective, we empirically compared the performance of microsatellites and SNPs to infer the relationship among breeds, considering datasets of sheep and cattle from the literature. For each breed, genotypes were extracted from studies conducted with microsatellites and from studies performed with SNP chips. The datasets were chosen to consider limited geographical scales (mainly national) so as to be able to conduct fine genetic analyses. The conclusions of this study may lead to reconsider the possible interest toward microsatellites in current and future management of FAnGR.

## 2. Materials and Methods

### 2.1. Datasets

We studied three pairs of datasets including (i) five local Algerian sheep breeds, (ii) 17 French sheep breeds (mostly local with one composite), and (iii) seven local French cattle breeds. All datasets were available online except the microsatellite dataset for French sheep [21]. For all the datasets, the sampling aimed to limit kinship as much as possible, using pedigree when available and/or sampling animals from as many different birth flocks as possible. As far as possible, the datasets were balanced, i.e., 20 individuals were considered for each population. The individual inbreeding levels (F_IS_) obtained with PLINK v1.07 [22] and Genepop v4.7 [23,24] software for SNPs and microsatellites, respectively, were used as criteria in the selection of individuals to be retained. All details concerning the datasets are shown in Table 1 and Appendix A.

#### 2.1.1. Microsatellites Datasets

The considered datasets included between 21 and 30 microsatellites (see details in Table 1). Most microsatellites, regardless of the dataset, were selected according to ISAG/FAO recommendations [25]. The list of the microsatellites used in each dataset is shown in Appendix A. Details on primers, original references and experimental protocols (conditions of PCR, multiplexing) can be found in: [26] for the French cattle dataset; [27] for the Algerian sheep dataset; and [21] for the French sheep dataset.

Polymorphic information content (PIC), observed heterozygosity (H_O_) and effective number of alleles (A_e_) were estimated for all markers using Molkin software (version 2.0) [28].

#### 2.1.2. SNPs Datasets

Cattle genotypes [29] were obtained using the Illumina BovineSNP50v1 BeadChip. Algerian sheep genotypes [30,31] were obtained using the Illumina OvineSNP50K BeadChip. French sheep population genotypes [32] were obtained using the Illumina Ovine HD SNP chip; for the purpose of the study, a subset of SNPs was extracted using the OvineSNP50K BeadChip coordinates. The following filtering parameters were applied with PLINK: (i) SNP call rate ≤ 97%, (ii) SNP minor allele frequency (MAF) ≤ 10%, (iii) animals with ≥10% missing genotypes, and (iv) SNPs that did not pass the HWE test (*p* ≤ 0.001).

### 2.2. Data Analysis

We computed Nei’s [33] pair-wise F_ST_ values and the associated 95% confidence intervals using the Hierfstat package [34] available for R version 3.5.0 [35]. Following the recommendations of Malomane et al. [36], no linkage disequilibrium (LD)-based SNP pruning was performed prior to the F_ST_ computations. NeighborNet graphs based on Reynolds’ [37] genetic distances were constructed using SplitsTree [38].

To ensure that uncorrected LD did not distort the analysis, SNP pruning was used to identify a subset of SNPs using the --indep option of PLINK with the following settings: 50 SNPs per window, a shift of five SNPs between windows, and a variation inflation factor threshold of two (corresponding to r^2^ > 0.5).

Genetic clusters of individuals were identified via a Bayesian model-based approach implemented in STRUCTURE 2.3.4 [39,40,41] using: an admixture model, correlated allele frequencies, 50,000 burn-in followed by 200,000 simulations. Convergence was checked using ten runs for each K value. The most probable value of K was estimated on the basis of the ΔK statistic [42]. The program CLUMPAK [43] available at http://clumpak.tau.ac.il, was used to analyze the multiple independent runs at a single K value and to visualize the results.

## 3. Results and Discussion

Microsatellite datasets included 113 (Algerian sheep dataset) to 425 (French sheep dataset) individuals (Table 1). The total number of alleles varied from 283 (French cattle dataset) to 343 (Algerian sheep dataset) (Table 1). The informative significance of the different microsatellite datasets was substantial (see indices, mean A, mean PIC, mean H_O_ and mean A_e_).

The SNP datasets included 36 (Algerian sheep dataset) to 346 individuals (French sheep dataset). The number of SNPs available after filtration ranged from 36,493 (Algerian sheep dataset) to 47,286 (French cattle dataset) (Table 1).

Mean values of F_ST_ obtained from microsatellite datasets were very close to those obtained from SNP datasets (Table 1). Pearson correlation coefficient between pairwise F_ST_ values obtained for the microsatellite dataset and the SNP dataset ranged from 0.77 (French cattle dataset) to 0.95 (French sheep dataset). All correlation coefficients were significantly different from zero (Table 1). By considering in detail the 95% confidence intervals for each pair (values obtained from microsatellite and SNP datasets) of pairwise F_ST_ values, the following was observed: for the French sheep dataset, four pairs of confidence intervals were non-overlapping considering the 136 pairs available (Appendix A). For the Algerian sheep dataset, out of the 10 pairs of F_ST_ values, one showed non-overlapping confidence intervals (Appendix A). For the French cattle dataset, one pair on the 21 pairs of F_ST_ values showed no overlapping (Appendix A).

NeighborNet graphs, based on Reynolds genetic distances, obtained from microsatellite genotyping or SNP genotyping gave highly coherent pictures of the breed relationships for each dataset (Figure 1, Figure 2 and Figure 3). The major difference was overestimation of genetic distance between Ouled-Djellal and Rembi in the Algerian dataset (Figure 1), by microsatellites compared to results obtained with SNP. The same situation occurred for Salers and Aubrac (cattle dataset, Figure 2). The French sheep dataset, comprising 17 breeds, requires more commentary (Figure 3). Both analyses correctly ordered the breeds according to their location of origin. Moreover, they clearly distinguished the breeds of the Massif Central/South of France from the others. In the same way, the genetic peculiarity of Mérinos de Rambouillet, a patrimonial breed maintained as a closed flock for around 230 years [21], was obvious. The Romane, a recent composite breed, appeared grouped with the Berrichon du Cher, an expected result given the breed was developed by crossing Romanov (not included in the dataset) with Berrichon du Cher. Differences were mainly seen in the cluster including North-West breeds (i.e., Roussin de la Hague, Rouge de l’Ouest) and Charmoise. This cluster appeared quite far from the cluster including Berrichon du Cher/Romane in the microsatellite analysis whereas the SNP analysis emphasized proximity for these two clusters versus the breeds of the Massif Central/South of France. Charmoise was developed in the middle of the 19th century by crossing local breeds (i.e., Berrichon du Cher, Solognot (not included in the dataset), and Mérinos de Rambouillet) with Romney sheep (not included in the dataset) imported from the United Kingdom. Moreover, Roussin de la Hague, Rouge de l’Ouest and Berrichon du Cher were largely subjected to the influences of English sheep breeds, to which is added the substantial influence of Merinos de Rambouillet for the Berrichon du Cher breed. The history of these breeds explains their grouping, which was only captured as a whole by the SNP analysis, whereas the microsatellite analysis was more sensitive to the strong link between Berrichon du Cher and Merinos de Rambouillet.

STRUCTURE analyses also showed very consistent patterns between the two types of markers (Figure 4, Figure 5 and Figure 6). In general, patterns obtained from microsatellite datasets were noisier than those from SNP datasets.

Considering the French sheep dataset (Figure 4), the most likely number of clusters according to the ΔK criterion was K = 3 for the microsatellite dataset, highlighting the existence of the three groups: (i) the Massif Central/South of France group, (ii) the Charmoise/Rouge de l’Ouest/Roussin de la Hague group, and (iii) the Berrichon/Romane/Mérinos de Rambouillet group. For the SNP analysis, the ΔK criterion identified K = 2 as the most likely number of clusters, highlighting Mérinos de Rambouillet as the major distinction versus the other breeds. Both analyses underlined the clear distinction between the Massif Central/South of France breeds and the others (K3 and K4). Considering the Massif Central/South of France group, both analyses emphasized the genetic peculiarity of Manech Tête Rousse, the proximity of Noire du Velay/Rava/Causse du Lot, as well as of Lacaune meat/Lacaune milk, and also Blanche du Massif Central/Préalpes du Sud/Moureros/Tarasconnaise. The major distinctions were as follows: for K = 2, the microsatellites clustered Charmoise/Roussin de la Hague/Rouge de l’Ouest/Berrichon/Romane/Mérinos de Rambouillet. Then, at K = 3, they postulate a split within this group, with Charmoise/Rouge de l’Ouest/Roussin de la Hague on one hand and Berrichon/Romane/Mérinos de Rambouillet on the other. The SNP analysis, on the contrary, grouped all these breeds together with the exception of the Mérinos de Rambouillet breed (K = 3). This difference was already discussed in relation with the results of the NeighborNet analysis (Figure 3).

Considering the Algerian sheep dataset (Figure 5), the most likely number of clusters according to the ΔK criterion was K = 3 for the microsatellite dataset, emphasizing the existence of the three groups: Sidaoun/D’Men, Ouled-Djellal/Rembi and Hamra. For the SNP analysis, K = 4 was identified as the most likely number of clusters. At this K value, all breeds were discriminated by both methods except the admixed Ouled-Djellal and Rembi breeds. Moreover, a clear link appeared between the Sidaoun and D’Men breeds. Hence, the two figures show a similar picture of the Algerian genetic structure.

Considering the French cattle dataset (Figure 6), the most likely number of clusters according to the ΔK criterion was K = 2 for both analyses. At this K value, the peculiarity of Rouge des Prés and its link with Bretonne Pie Noire and Charolais were highlighted. Both analyses were highly correlated and the main distinction was observed for Aubrac and Charolais, which show a clearer admixture with the SNP analysis (K = 7) than with microsatellites.

This study compared genetic structures derived from microsatellites to that inferred by SNPs, considering three pairs of datasets for two species of domesticated mammals.

Population genetic differentiation analyses, as well as STRUCTURE analyses showed a very strong consistency between the two types of markers. In particular, the French sheep dataset, including 17 breeds shaped by a rich history (English and Spanish influences among others), made it possible to fully appreciate the usefulness of microsatellite data. It must be noted that, for each pair of datasets (whatever the species considered), it is not the same individuals in each breed who have been genotyped with microsatellites and SNPs. Thus, it is quite remarkable to find such a correlation in the obtained patterns, while the variability introduced by this “sampling effect” constitutively limits the overlap of results from the beginning.

Faced with the power of SNP chips, which are able to capture even tenuous relationships between breeds and to estimate confidence intervals with the highest precision, etc., it turns out that microsatellites gave pictures that were largely concordant, although with less accuracy. Microsatellite markers were able to capture genetic patterns (gene flows, admixture, etc.) considering national scales, which is one of the first requirements for defining and prioritizing conservation measures.

According to the Food and Agriculture Organization (FAO) [1], the risk status for more than 80% of mammalian breeds in Africa and Latin America is unknown. These alarming figures reveal the extent to which ignorance about livestock is high in most developing countries. When the number of head per flock is often not even known, the implementation of expensive genetic analyses is difficult to envisage. In such a context, the use of microsatellites seems to be the most appropriate solution for various reasons. (i) The cost associated with microsatellite genotyping is significantly lower than that of the SNP chip. Indeed, commercially available kits (e.g., the Bovine Genotypes Panel 3.1, the Equine Genotypes Panel 1.1 Kit, the Canine ISAG STR Parentage Kit 2014) provide reagents for the genotyping of about twenty multiplexed microsatellites for an average sum of 13 USD per individual. Moreover, in-house genotyping may substantially reduce this price (i.e., approximately 7 USD per individual). For sheep, goat, pig and cattle the genotyping cost, via Beadchip of around 50K SNP markers, is 67 USD/individual on average, depending on the species and the manufacturer. (ii) High throughput genotyping techniques generate huge data files, the analyses of which require a powerful computing cluster and associated computer knowledge [44]. (iii) Finally, it should be stressed that breeds are still largely managed in the traditional way in developing countries. In order to understand the genetic pattern at a national scale and to answer very practical concerns such as the assessment of introgression and indiscriminate cross-breeding within local populations [45], a broad sample should be undertaken first. Indeed, sampling must be sufficiently broad to cover the global range of the breeds. Only this type of sampling will make it possible to really grasp the situation, knowing that a whole range of situations can be encountered between the preserved populations and the diluted populations [46]. The current cost of genetic chips is hardly compatible with the need for broad sampling, and once again emphasizes the appeal of using microsatellites to provide a first rough, but broad picture of the national situation (see [47]).

It should also be noted that microsatellite kits are still widely used in the private sector for paternity testing. As a result, there is a significant amount of data, especially for cattle and horses, that could be interesting to collect and analyze as part of studies focusing on genetic diversity.

By publishing microsatellite sets for different species of domestic animals [25], the FAO, has created an interesting dynamic in this context. For example, the use of common sets in different studies allows a comparison, although partial, between the datasets Indeed, data sets can be merged only when standardized protocols are followed (i.e., ISAG/ICAR standards) and laboratories use the same microsatellite panels and share a few individuals, playing the role of control, in common. The use of SNP chips, which implies a higher cost as well as the possession of associated computer platforms, could occur and be beneficial at the second stage in order to refine the conclusions, to push the studies towards much finer research (e.g., the search for selection signatures), or given the ease of merging, to compare datasets from different countries.

## 4. Conclusions

In conclusion, this study highlights the role that microsatellites still play in the management of FAnGR. Indeed, our results show that microsatellites provide a picture of the genetic structure, and even though it is less accurate than that obtained with SNPs, it is absolutely relevant. Therefore, microsatellites are a suitable tool to make an initial evaluation of the situation. Because the effects of climate change are becoming more pressing, it does not seem appropriate to wait until access to high throughput techniques is widespread to characterize the breeds of developing countries. These breeds, which are largely adapted to their environments, constitute an invaluable heritage that can help us to face climate changes [48].

## Figures and Tables

**Figure 1 genes-11-00057-f001:**
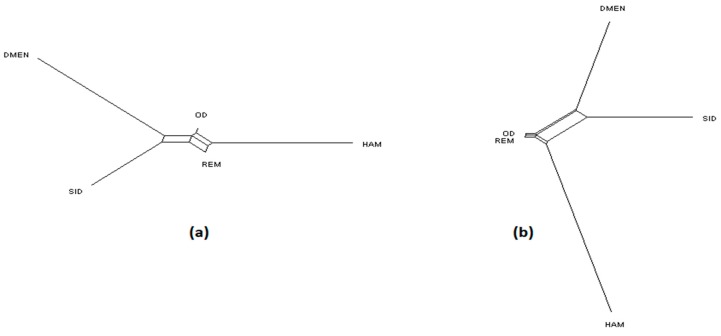
NeighborNet graph considering Algerian sheep breeds, from a matrix of Reynolds’ distances. (**a**) Plot obtained from the microsatellite dataset; (**b**) plot obtained from the single nucleotide polymorphism (SNP) dataset. For breed names see codes in Appendix A.

**Figure 2 genes-11-00057-f002:**
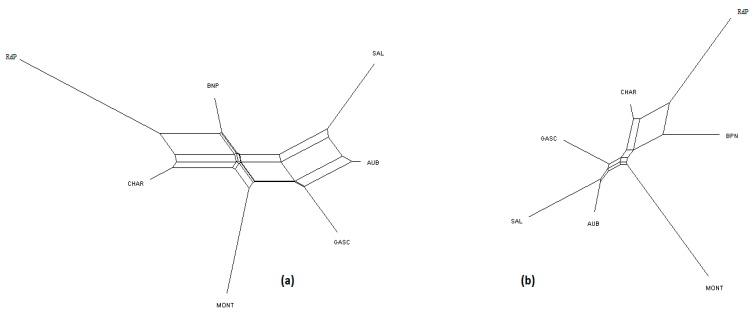
NeighborNet graph considering French cattle breeds, from a matrix of Reynolds’ distances. (**a**) Plot obtained from the microsatellite dataset; (**b**) plot obtained from the SNP dataset. For breed names see codes in Appendix A.

**Figure 3 genes-11-00057-f003:**
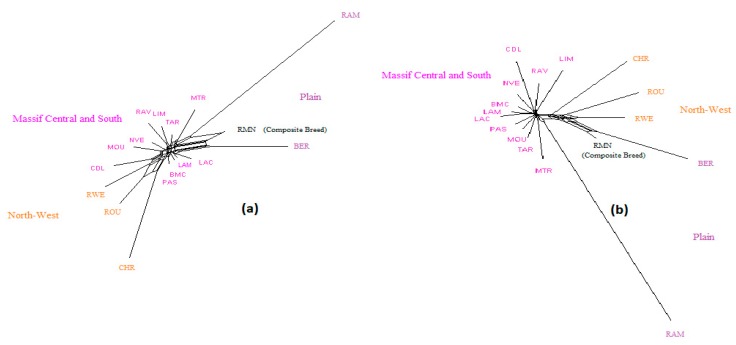
NeighborNet graph considering French sheep breeds, from a matrix of Reynolds’ distances. (**a**) Plot obtained from the microsatellite dataset; (**b**) plot obtained from the SNP dataset. Information concerning the region of origin of each breed was extracted from [21]; for breed names see codes in Appendix A.

**Figure 4 genes-11-00057-f004:**
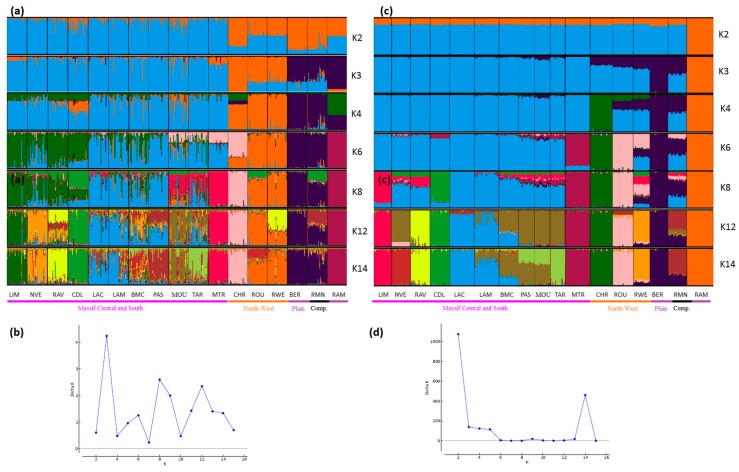
Genetic structure of French sheep breeds by Bayesian analysis (K = number of clusters). (**a**) STRUCTURE plot obtained from the microsatellite dataset; (**b**) graph showing ΔK calculated according to [42] for the microsatellite dataset; (**c**) STRUCTURE plot obtained from the SNP dataset; (**d**) graph showing ΔK calculated according to [42] for the SNP dataset. Comp.: composite breeds; information concerning the region of origin of each breed was extracted from [21]; for breed names see codes in Appendix A.

**Figure 5 genes-11-00057-f005:**
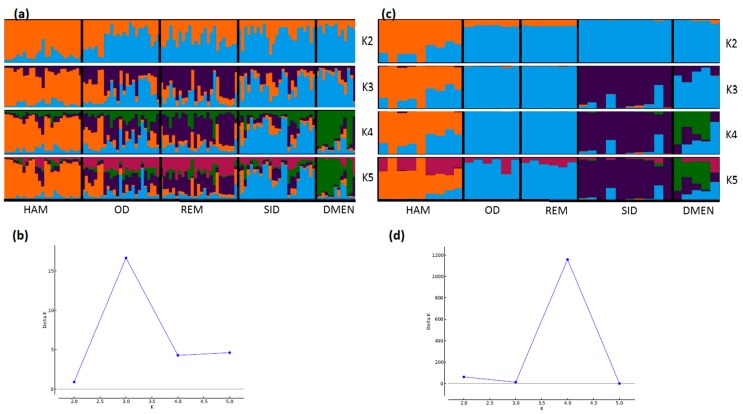
Genetic structure of Algerian sheep breeds by Bayesian analysis (K = number of clusters). (**a**) STRUCTURE plot obtained from the microsatellite dataset; (**b**) graph showing ΔK calculated according to [42] for the microsatellite dataset; (**c**) STRUCTURE plot obtained from the SNP dataset; (**d**) graph showing ΔK calculated according to [42] for the SNP dataset. For breed names see codes in Appendix A.

**Figure 6 genes-11-00057-f006:**
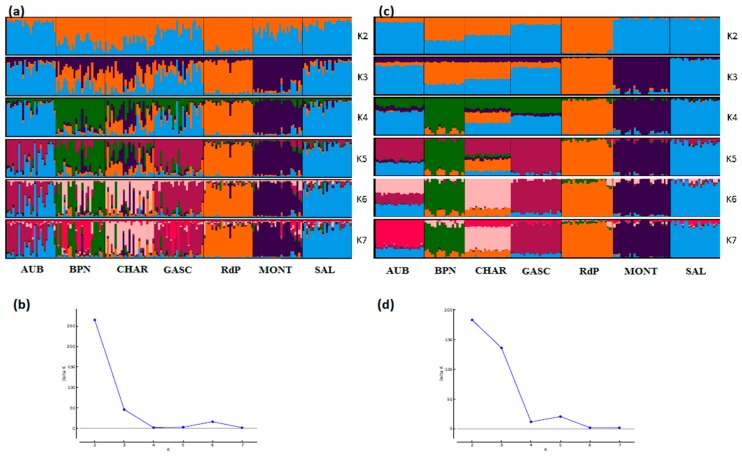
Genetic structure of French cattle breeds by Bayesian analysis (K = number of clusters). (**a**) STRUCTURE plot obtained from the microsatellite dataset; (**b**) graph showing ΔK calculated according to [42] for the microsatellite dataset; (**c**) STRUCTURE plot obtained from the SNP dataset; (**d**) graph showing ΔK calculated according to [42] for the SNP dataset. For breed names see codes in Appendix A.

**Table 1 genes-11-00057-t001:** Genetic diversity measured by dataset.

	Algerian Sheep Datasets	French Sheep Datasets	French Cattle Datasets
no. of breeds	5	17	7
Microsatellite datasets:			
nb. of individuals	113	425	175
nb. of microsatellites	29	21	30
nb. of alleles	343	292	283
mean A (s.d.)	11.83 (13.29)	13.90 (21.59)	9.43 (11.97)
mean H_O_ (s.d.)	0.77 (0.008)	0.73 (0.018)	0.71 (0.015)
mean PIC (s.d.)	0.74 (0.010)	0.70 (0.018)	0.67 (0.018)
mean A_e_ (s.d.)	5.06 (4.67)	4.46 (3.89)	3.98 (3.02)
SNP datasets:			
nb. of individuals *	36	346	152
nb. of SNP	52,412	40,454	52,324
nb. of SNP after filtration	36,493	39,800	47,286
nb. of SNP after Pruning **	15,560	31,184	24,841
Mean F_ST_ from microsatellites datasets (s.d.)	0.048 (<0.001)	0.104 (0.004)	0.076 (<0.001)
Mean F_ST_ from SNP datasets (s.d.)	0.048 (<0.001)	0.105 (0.004)	0.078 (<0.001)
r Pearson *** (*p*-value)	0.87 (0.001)	0.95 (<0.001)	0.77 (<0.001)

no.: number; s.d.: standard deviation; A: number of alleles; H_O_: observed heterozygosity; PIC: polymorphic information content; Ae: effective number of alleles; *: after filtration (see Material and Methods); **: see Material and Methods; ***: Pearson correlation coefficient between pairwise F_ST_ values obtained with the microsatellite dataset and the SNP dataset.

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
