# Peer review of "Inference of Breed Structure in Farm Animals: Empirical Comparison between SNP and Microsatellite Performance"

_genes, 2020, doi:10.3390/genes11010057_

Round 1

Reviewer 1 Report

In my opinion list of STR in TableS2.docx (which I asked for) should be in that order for sheep, that reader could be found in easy way, which STR are common for Algerian and French sheep. Moreover, authors didn't give such facilitation as common order and didn't care about normalization of STRs names - e.i. ILST5 and ILSTS005, MAF214 and MAF0214, OAP34 and OARCP34, and so on - those pairs of STR in two sets of dataset are probably the same STRs. It should be clear in TableS2.docx which STRs are the same or not.

Author Response

In my opinion list of STR in TableS2.docx (which I asked for) should be in that order for sheep, that reader could be found in easy way, which STR are common for Algerian and French sheep. Moreover, authors didn't give such facilitation as common order and didn't care about normalization of STRs names - e.i. ILST5 and ILSTS005, MAF214 and MAF0214, OAP34 and OARCP34, and so on - those pairs of STR in two sets of dataset are probably the same STRs. It should be clear in TableS2.docx which STRs are the same or not.

Our answer: Thank you for once again agreeing to consider our work. We have modified Table S2 according to your instructions: (i) the names are in alphabetical order, (ii) we have carefully checked that the same microsatellites were named in the same way. In addition, we have bolded the microsatellites used in different studies to make this information clearer.

Microsatellites used in the Algerian sheep dataset

Microsatellites used in the French sheep dataset

Microsatellites used in the French cattle dataset

BM1824

BM8125

BM1818

BM8125

CSRD247

BM1824

CSRD247

HSC

BM2113

CSSM66

HUJ616

CSRM60

DYMS1

ILST11

CSSM66

HSC

ILSTS5

ETH10

HUJ616

INRA49

ETH152

ILST11

MAF214

ETH185

ILSTS5

MAF209

ETH225

INRA35

MAF65

ETH3

INRA63

MAF70

HAUT24

INRA49

MCM42

HAUT27

MAF209

MCM527

HEL13

MAF214

OARCP34

HEL13

MAF33

OARFCB128

HEL5

MAF65

OARFCB193

HEL9

MCM140

OARFCB20

ILSTS5

MCM42

OARFCB304

ILSTS6

MCM527

OARJMP29

INRA23

OARAE129

OARJMP58

INRA32

OARCP34

SRCRSP9

INRA35

OARFCB128

INRA37

OARFCB193

INRA5

OARFCB20

INRA63

OARFCB304

MM12

OARJMP58

SPS115

SRCRSP9

TGLA122

TGLA122

TGLA126

TGLA53

TGLA227

TGLA53

In bold, microsatellites used in at least two different studies

Reviewer 2 Report

The manuscript has been significantly improved and I suppose, that it will be quite interesting for the readers.

I have still one question and I recommend only the minor revisions:

1. In Belabdi et al (2019) there were 47 Algerian samples. In present paper only 36 samples were used. Were eleven animals excluded due to their close relations?

2. I recommend to rewrite and to expand the Conclusions.
The authors should emphasize the observed results on the more or less comparable power of microsatellites with SNP data to provide an insight the primary population structure and genetic relations of the livestock.

Author Response

The manuscript has been significantly improved and I suppose, that it will be quite interesting for the readers.

Our answer: Thanks for allowing us to improve the manuscript and for agreeing to consider our revisions once again.

I have still one question and I recommend only the minor revisions:

In Belabdi et al (2019) there were 47 Algerian samples. In present paper only 36 samples were used. Were eleven animals excluded due to their close relations?

Our answer: Relative to the sampling carried out for the study by Belabdi et al. 2019, a number of individuals could not be retained for this comparative analysis for the following reasons. (i) Two populations of the Hamra breed had been sampled for the Belabdi’s study (individuals from state farms and individuals from private farms), but in the microsatellite study only the population from the state farms had been analysed. Therefore, we could not consider the population from private farms of the Belabdi’s study because we did not have any point of comparison in microsatellites. (ii) In addition, many individuals of the Sidaoun breed had been sampled for the Belabdi’s study, but in order to balance the sample sizes, in the SNP analysis, we had to limit the number of Sidaoun individuals to be included.

I recommend to rewrite and to expand the Conclusions.
The authors should emphasize the observed results on the more or less comparable power of microsatellites with SNP data to provide an insight the primary population structure and genetic relations of the livestock.

Our answer: As recommended we have developed the conclusion by strengthening the comparison between SNPs and microsatellites in order to highlight the potential interest of microsatellites. The old version of the conclusion and then the modified version are shown below.

Old version:

In conclusion, this study highlights the interest that microsatellites retain in the management of FAnGR. Indeed, while the effects of climate change are becoming more pressing, it does not seem relevant to wait until access to high throughput techniques is widespread, to characterize the breeds of developing countries. These breeds, which are largely adapted to their environments, constitute an invaluable heritage that can help us face to climate changes [48].

Improved version:

In conclusion, this study highlights the interest that microsatellites retain in the management of FAnGR. Indeed, our results show that microsatellites provide a picture of the genetic structure, although less accurate than that obtained with SNPs, absolutely relevant. It is therefore a suitable tool to make an initial evaluation of the situation. While the effects of climate change are becoming more pressing, it does not seem appropriate to wait until access to high throughput techniques is widespread, to characterize the breeds of developing countries. These breeds, which are largely adapted to their environments, constitute an invaluable heritage that can help us face to climate changes [48].

This manuscript is a resubmission of an earlier submission. The following is a list of the peer review reports and author responses from that submission.

Round 1

Reviewer 1 Report

In general, the subject of the manuscript by Abbas Laoun et al. is of some scientific interest, however the idea is not original.

For example, Gärke C. et al. (Anim Genet. 2012, 43(4):419-28. doi: 10.1111/j.1365-2052.2011.02284.x) performed the studies of eight chicken breeds to determine the number of SNPs needed to obtain the same differentiation power as with standard set of 29 microsatellites. They observed the comparable resolution of microsatellite based PCA plot using only 70 SNPs. Nevertheless, the used set of 29 microsatellites has not enough power to detect the admixture in several chicken populations.  Fernandez M.E. et al. (Genetics and Molecular Biology, 2013, 36(2):185-191) performed a comparison of the effectiveness of microsatellites and SNP panels for genetic identification, traceability and assessment of parentage in an inbred Angus herd and showed, that 2-3 SNPs per STR were needed to obtain an equivalent Q values and matching probabilities values. Cortés O. et al. (Livestock Science, 2019, 219:80–85. https://doi.org/10.1016/j.livsci.2018.11.006) observed only a moderate correlation values (0.43–0.54) among derived metrics, which were calculated based on 24 microsatellites and 50K SNP inbreeding coefficients at study of Lidia cattle breed. We gave only a few examples of works aimed at the comparison of resolution power of microsatellites and SNPs for study of parentage, population structure and genetic diversity in domestic animal species. I suppose that such kind of papers should be at least cited and discussed at the present manuscript. The results are well described, and the data was correctly processed, however I have an impression, that authors only applied the broad range of conventional statistic methods to microsatellite and SNP data without any specific goal. I suggest, that authors should find the concrete objective which distinguishes their study from the similar works.

There are a few other points that should be clarified.

Major comments

Is there some input information on the animals included in the datasets regarding their kinship? Were the animals from both STR and SNP dataset checked by their relatedness? The biases in the data interpretation could be occurred in case the animals from the SNP dataset were tested and close relatives were excluded, but the animals from the STR dataset were not checked and relatives are remained. Have the authors considered these issues? Why the number of individuals used for microsatellite and SNP studies is differed? For correct comparison it would be necessary to use the same individuals genotyped for microsatellites and SNPs. If the number of individuals used for SNP-based analysis was lower due to exclusion of the part of individuals based on the quality control results, I suggest to exclude from the microsatellite data set the individuals which passed not through the quality control of SNP data. The both of data sets used for microsatellites and SNP studies should include the same individuals. 12 L. 297-301. The explanations on the problems in the interpretation of the porcine data sets are very ambiguous. What was the reason of the using the data on the wild boars if was out of placed and possibly created some misunderstandings? I suggest to exclude this data set from the study or to recalculate the data with excluding the wild boar from the data set. The use of join data set can lead to the biased results of genetic structure within domestic pigs’ breeds due to the long genetic distances between domestic swine and wild boar (see Lenstra J.A. et al., Animal Genetics. 2012; 43(5): 483-502). The authors are not consequent at description of results and discussion. The main idea is not tracked throughout the text. And the discussion almost repeats the results. The main idea of this paper at its present version, as I understood, is to show, that the using the microsatellites allows to obtain the similar results with SNPs at characterizing the genetic structure and genetic diversity of populations at different domestic animal species. But in discussion section authors points on the possibility of using low-density chips. It feels like the authors are just tired. The Discussion section is vague and comprises the re-written results.

Minor comments

3 L. 96-97: Why were STR loci selected according to FAO recommendations, but not ISAG? 3 L. 133-134: It is not correct to compare the variation in the number of microsatellite alleles between different data sets, because the different microsatellites were used for each species. Please, correct the abbreviations of breeds at Fig. 3 (“mont” at Fig. 3A but “mon” at Fig. 3B) and Fig. 4 (“cmh” at Fig. 4A but “chm” at Fig. 4B). I recommend using the English improving services.

Reviewer 2 Report

96-97: Most microsatellites, regardless of the dataset, were selected according to FAO recommendations

List of STR markers for all analysis is required - at least species and name of markers for this species. "Most [...] selected according to FAO" is not precision. Because of lack of this information, results (statistics) from this paper cannot be use in discussion (as citation) in further papers. Comparison of statistic values make sense only if analyses are done with the same molecular tool and STR panel

132-133: The microsatellites used in the different datasets showed high overall polymorphism.

Generalization, not precise. Table 1 contain only mean PIC. What authors have meant "high overall polymorphism". High polymorphism is above 0.7. Markers with above should be listed or amount those markers per markers in panel should be known.

What PIC forumla using Molkin software? PIC value should be like zero-point-number (0.754 for example or 1.000 for monomorphic markers). But in Table 1 there for example 74.06.

Please provide explenations.

“Results”

“Results” contain historical background (on page 5) which is not result of research. Title “Result” should be changed (example “Results and Discussion”) or historical background should be should be in “Discussion”

344-347: interesting dynamic in this context. The use of common sets between the different studies even allows a comparison, although partial, between the datasets. Indeed datasets can be merged, only, in very specific cases (e.g. analyses carried out by the same laboratory for example).

No, sentence in bracket does not have sense or could be understand by reader in correct way. STR data could be merged when there are standardized (e.i. ISAG, ICAR standards) and laboratories using the same sets of markers.

shortcut of observed heterozygosity (Ho) has many forms – this is correct (or other, but one constantly)

meanwhile there are incorrect forms

101: observed heterozygosity (HO)

138 datasets considering mean HO

In Table 1

143: HO: observed heterozygosity